# EgoVid-5M: A Large-Scale Video-Action Dataset for Egocentric Video Generation

Xiaofeng Wang[1,2]    Kang Zhao[1]    Feng Liu[4]    Jiayu Wang[1]
Guosheng Zhao[2,3]    Xiaoyi Bao[2,3]    Yingya Zhang[1]    Xingang Wang[2]    Zheng Zhu[2†]
[1] Tongyi Lab,    [2] Tsinghua University,    [3] CASIA,    [4] UCAS
https://egovid.github.io/

## Abstract

Video generation has emerged as a promising tool for world simulation, leveraging visual data to replicate real-world environments. Within this context, egocentric video generation, which centers on the human perspective, holds significant potential for enhancing applications in virtual reality, augmented reality, and gaming. However, the generation of egocentric videos presents substantial challenges due to the dynamic nature of egocentric viewpoints, the intricate diversity of actions, and the complex variety of scenes encountered. Existing datasets are inadequate for addressing these challenges effectively. To bridge this gap, we present *EgoVid-5M*, the first high-quality dataset specifically curated for egocentric video generation. *EgoVid-5M* encompasses 5 million egocentric video clips and is enriched with detailed action annotations, including 5M high-level textual descriptions and 65K fine-grained kinematic control annotations. To ensure the integrity and usability of the dataset, we implement a sophisticated data cleaning pipeline designed to maintain frame consistency, action coherence, and motion smoothness under egocentric conditions. Furthermore, we introduce *EgoDreamer*, which is capable of generating egocentric videos driven simultaneously by action descriptions and kinematic control signals. The *EgoVid-5M* dataset, associated action annotations, and all data cleansing metadata will be released for the advancement of research in egocentric video generation.

## 1 Introduction

One of the most promising avenues in video generation is the development of world simulators, which utilize visual simulations and interactions to deliver applications in the physical world. Contemporary research is increasingly validating such capabilities, including applications in autonomous driving [1, 2, 3, 4, 5, 6], autonomous agents [7, 8, 9, 10, 11, 12, 13], and even in general world [14, 15]. In human-centric scenarios, leveraging behavioral actions to drive egocentric video generation has emerged as a pivotal strategy. This approach enhances applications in Virtual Reality (VR), Augmented Reality (AR), and gaming, offering immersive and interactive experiences and advancing the state of the art in these fields.

Video generation requires large amounts of high-quality data, especially for egocentric videos, which are challenging due to their dynamic nature, rich actions, and diverse scenarios. Currently, there is a significant lack of large-scale, specialized datasets for training egocentric video generation models. To address this, we introduce the *EgoVid-5M* dataset, a high-quality resource specifically curated for egocentric video generation. As shown in Tab. 1, *EgoVid-5M* is distinguished by several key features: (1) **High Quality**: It consists of 5M 1080p egocentric videos, rigorously cleaned to ensure alignment

---

†Corresponding author. zhengzhu@ieee.org

| Dataset | Year | Domain | *Gen.* | Text | Kinematic | *CM.* | #Videos | #Frames | Res |
|---|---|---|---|---|---|---|---|---|---|
| HowTo100M [19] | 2019 | Open | ✓ | ASR | ✗ | ✗ | 136M | $\sim 90$ | 240p |
| WebVid-10M [20] | 2021 | Open | ✓ | Alt-Text | ✗ | ✗ | 10M | $\sim 430$ | Diverse |
| HD-VILA-100M [21] | 2022 | Open | ✓ | ASR | ✗ | ✗ | 103M | $\sim 320$ | 720p |
| Panda-70M [22] | 2024 | Open | ✓ | Auto | ✗ | ✗ | 70M | $\sim 200$ | Diverse |
| OpenVid-1M [23] | 2024 | Open | ✓ | Auto | ✗ | ✗ | 1M | $\sim 200$ | Diverse |
| VIDGEN-1M [24] | 2024 | Open | ✓ | Auto | ✗ | ✗ | 1M | $\sim 250$ | 720p |
| LSMDC [25] | 2015 | Movie | ✗ | Human | ✗ | ✗ | 118K | $\sim 120$ | 1080p |
| UCF101 [26] | 2015 | Action | ✗ | Human | ✗ | ✗ | 13K | $\sim 170$ | 240p |
| Ego4D [27] | 2022 | Egocentric | ✗ | Human | IMU | ✗ | 931 | $\sim 417K$ | 1080p |
| Ego-Exo4D [28] | 2024 | Egocentric | ✗ | Human | MVS | ✗ | 740 | $\sim 186K$ | 1080p |
| **EgoViD-5M (ours)** | 2024 | Egocentric | ✓ | Auto | VIO | ✓ | 5M | $\sim 120$ | 1080p |

Table 1: Comparison of *EgoVid-5M* and other video datasets, where *Gen.* denotes whether the dataset is designed for generative training, *CM.* denotes cleansing metadata, #Videos is the number of videos, and #Frames is the average number of frames in a video.

between action descriptions and video content, and consistent frame quality. (2) **Comprehensive Scene Coverage**: The dataset covers a wide range of scenarios, including household environments, outdoor settings, office activities, sports, and skilled operations, with hundreds of action categories. (3) **Detailed and Precise Annotations**: It includes fine-grained kinematic control and high-level action descriptions. Kinematic information is annotated using Visual Inertial Odometry (VIO), while action descriptions are generated by a multimodal large language model.

Leveraging the proposed *EgoVid-5M*, we train different video generation baselines to validate the dataset's quality and efficacy, including U-Net [16, 17] and DiT [18]) models. Experimental results demonstrate that *EgoVid-5M* bolsters the training of egocentric video generation. In addition, we propose *EgoDreamer*, which utilizes both action descriptions and kinematic control to drive the generation of egocentric videos. To provide a comprehensive assessment of action-driven egocentric video generation, we establish a set of evaluation metrics. These metrics encompass multiple dimensions, including visual quality, frame coherence, semantic compliance with actions, and kinematic accuracy. Extensive experiments show that *EgoVid-5M* enhances the capability of various video generation models to produce high-quality egocentric videos.

The main contributions of this paper can be summarized as follows: (1) We introduce *EgoVid-5M*, the first publicly released, high-quality dataset tailored for egocentric video generation. This dataset is proposed to advance both research and applications in the domain of egocentric visual simulation. (2) Our dataset includes detailed and precise action annotations, incorporating both fine-grained kinematic control and high-level textual descriptions. In addition, we employ robust data cleaning strategies to ensure frame consistency, action coherence, and motion smoothness within *EgoVid-5M*. (3) Utilizing *EgoVid-5M*, we conducted extensive experiments on various video generation baselines to validate the dataset's quality and efficacy. Furthermore, to support future advancements in action-driven egocentric video generation, we propose *EgoDreamer*, which leverages both action descriptions and kinematic control to drive egocentric video generation.

## 2 Related Work

### 2.1 Video Generation as World Simulators

Video generation technology has seen rapid advancements recently. Both diffusion-based [29, 16, 30, 31, 17, 32, 33, 18] and token-based [34, 35, 36, 37, 38, 39] video generation models have proven that the quality and controllability of video generation are steadily improving [40]. Notably, the introduction of the Sora model [14] attracts significant attention which convincingly shows that current video generation models are capable of understanding and adhering to physical laws, thereby substantiating the potential of these models to function as world simulators. This perspective is echoed by Runway, which posits that their Gen-3 Alpha [41] is progressing along this promising trajectory. Additionally, video generation models, employed as simulators, have demonstrated significant utility in various real-world applications, including autonomous driving simulations [1, 3, 5, 4, 6, 2] and agent-based environments [7, 8, 9, 10, 11, 12, 13]. Within this context, action-

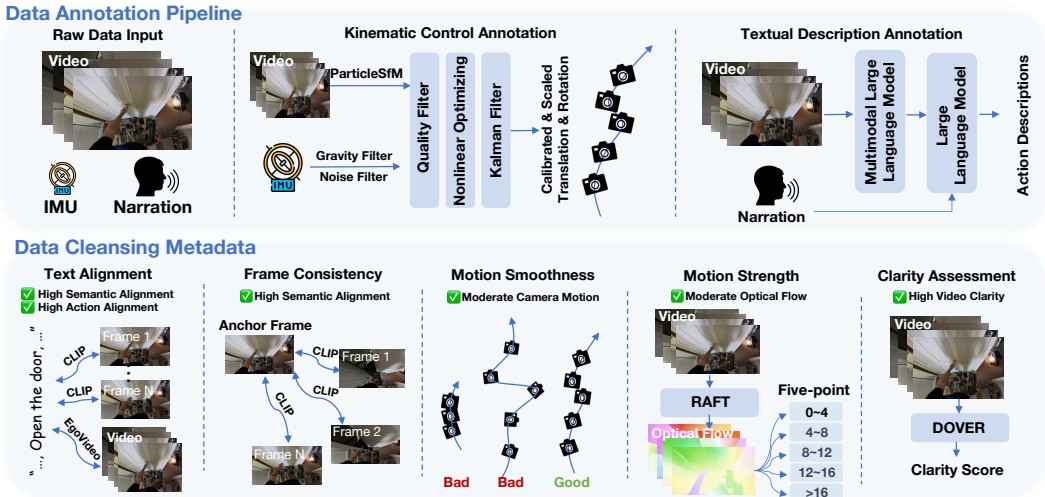

Figure 1: Data annotation pipeline and cleansing metadata of *EgoVid-5M*.

driven egocentric video generation, which centers on the human perspective, holds significant potential for enhancing applications in VR, AR, and gaming. However, current research in the egocentric domain predominantly concentrates on understanding tasks [42, 43, 44, 45, 46, 47, 48, 49, 50], and generative tasks associated with egocentric scenarios are largely confined to exocentric-to-egocentric video synthesis [51, 52, 53]. This highlights a substantial gap in generating action-driven egocentric videos. While some methods have explored video generation driven by action interaction [54, 55, 56, 57, 58, 59, 60, 61], these approaches are mainly concerned with natural scenes featuring smooth camera transitions. This focus limits their ability to model intricate motion patterns inherent in egocentric videos.

## 2.2 Video Generation Datasets

In the realm of video generation, the quantity and quality of training data are pivotal for training effective models. Currently, the field of general video generation benefits from several pioneering open-source video datasets. WebVid-10M [20] consists of 52K hours of video, totaling 10.7M text-video pairs. Similarly, InternVid [62] contains over 7M videos spanning nearly 760K hours, with 4.1B words in descriptive texts. Panda70M [22] stands out with its collection of 70M high-resolution and semantically coherent video samples. OpenVid-1M [23], offers a million-level, high-quality dataset encompassing diverse scenarios such as portraits, landscapes, cities, metamorphic elements, and animals. In contrast to these general-purpose datasets, specific-scenario datasets typically comprise a limited number of text-video pairs. UCF-101 [26] is an action recognition dataset featuring 101 classes and 13,320 total videos. Taichi-HD [63], a more focused collection, includes 2,668 videos capturing a single person performing Taichi. In egocentric video generation, existing datasets such as Ego4D [27] and Ego-Exo4D [28] are primarily designed for egocentric understanding tasks and often include excessive noisy camera motion, making them unsuitable for generative training. Additionally, EgoGen [64], a synthetic dataset, can not fully encapsulate the complex variations in real-world egocentric views. To address this gap, we introduce the *EgoVid-5M* dataset, a pioneering and meticulously curated collection designed explicitly for egocentric video generation. *EgoVid-5M* comprises 5M egocentric video clips with precise action annotations and cleansing metadata.

## 3 EgoVid-5M

The training of video generation relies on large-scale, high-quality video data. Therefore, we built *EgoVid-5M* based on the large-scale Ego4D dataset [27]. Notably, although Ego4D contains thousands of hours of egocentric videos, it is intended for egocentric perception and includes excessive noisy camera motion that is unsuitable for generative training. Additionally, the narration annotation in Ego4D is overly simplistic and lacks semantic consistency with frames. To address

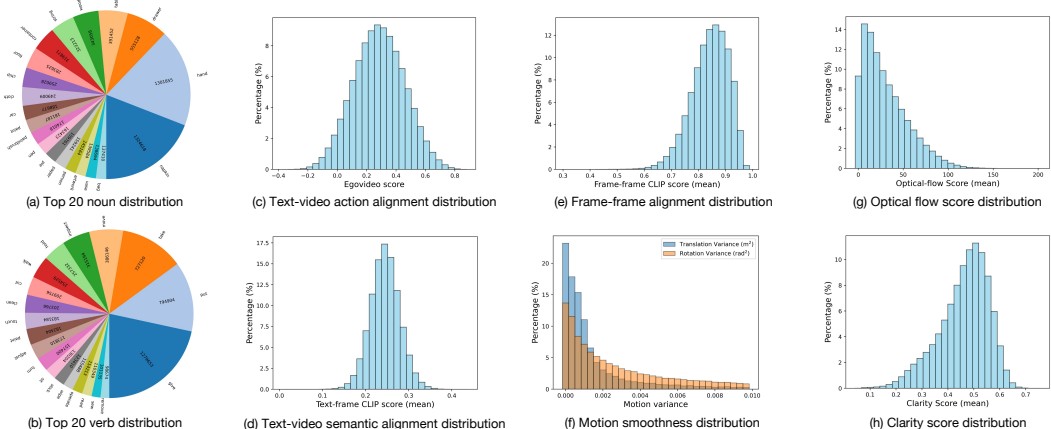

Figure 2: Data annotation distribution of *EgoVid-5M*. (a) and (b) describe the quantities of the top 20 verbs and nouns. (c) Text-video action alignment is assessed using the EgoVideo score. (d) and (e) measure the semantic similarity between text and frames and between frames and the first frame using the average CLIP score. (f) Motion smoothness is quantified by the variance of translation and rotation. (g) Motion strength is represented by the average global optical flow. (h) Video clarity is determined by the DOVER score.

these issues, we propose a data annotation pipeline that provides detailed and accurate annotations of fine-grained kinematic control and high-level action descriptions. Furthermore, a data cleaning pipeline is developed to ensure alignment between action descriptions and video content, as well as the magnitude of motion and consistency between frames.

## 3.1 Data Annotation Pipeline

In order to simulate egocentric videos from actions, we construct detailed and accurate action annotations for each video segment, encompassing low-level kinematic control (e.g., ego-view translation and rotation), as well as high-level textual descriptions. The annotation pipeline is shown in the upper part of Figure 1.

**Kinematic Control Annotation** In order to accurately describe complex egocentric movements, we utilize the Visual-Inertial Odometry (VIO) method to construct kinematic control signals. This involves using ParticleSfM [65] to obtain scale-ambiguous camera poses $P_c$ from video, followed by integrating IMU signals $\{I_t\}_{t=0}^{T-1}$ to obtain more accurate and scaled camera poses. However, there are several challenges to overcome. (1) The IMU signals are subject to noise. (2) The transformation matrix between the IMU and the camera is unknown. (3) The initial velocity of the IMU is unknown. (4) The scale factor of the $P_c$ is unknown. To address the aforementioned problems, we first utilize high-pass Butterworth filters $\mathcal{F}_{IFFT}(\mathcal{H}_{\text{low}}(s) \cdot \mathcal{F}(s))$ and low-pass Butterworth filters $\mathcal{F}_{IFFT}(\mathcal{H}_{\text{high}}(s) \cdot \mathcal{F}(s))$ to filter out the gravity signal and high-frequency noise, where $\mathcal{F}(s) = \mathcal{F}_{FFT}(I)$ is the *Fast Fourier Transform* and $\mathcal{F}_{IFFT}$ is the inverse operation. $\mathcal{H}_{\text{low}}(s) = \frac{1}{1+(\frac{s}{w_c})^{2n}}$ is the low-pass filter, $\mathcal{H}_{\text{high}}(s) = \frac{(\frac{s}{w_c})^{2n}}{1+(\frac{s}{w_c})^{2n}}$ is the high-pass filter, $w_c$ represents the cutoff frequency while $n$ represents the filter order. Next, we propose a quality filter to drop the low-quality $P_c$ and $I$, where the motivation is that the number of reconstructed points $N_p$ (generated from ParticleSfM) is a reflection of the accuracy of $P_c$ [56], and the variance of IMU reflects the dynamic nature of the video. Therefore, the retained data needs to simultaneously satisfy $N_p \geq N_{\text{thres}}$ and $\frac{1}{T}\sum_{t=0}^{T-1}(I_t - \bar{I})^2 \leq V_{\text{thres}}$. Next, we perform the least squares minimization with $P_c$ and the integrated IMU signal $\{I_t\}_{t=0}^{T-1}$ to calculate the initial velocity $v(0)$ of the IMU signal, the transformation matrix $T_I$ from IMU to the camera, and the scale factor $\lambda$ of the $P_c$:

$$\min_{v_0, T_I, \lambda} |T_I P_I(T-1) - \lambda P_c|^2, \tag{1}$$

where $P_I(T-1)$ can be derived from:

$$P_I(t+1) = P_I(t) + v(t)\Delta t + \frac{1}{2}I(t)\Delta t^2, \tag{2}$$

$$v(t+1) = v(t) + I(t)\Delta t, \tag{3}$$

with the initial condition $P(0) = \mathbf{0}$. Finally, we utilize the Kalman filter to fuse these two signals under the camera coordinate (see supplement for more details).

**Textual Description Annotation** In addition to kinematic control, another supplementary information of egocentric action is textual descriptions. In the Ego4D dataset, only human narrations serve as text annotations, but the narrations are relatively simple and lack semantic consistency with frames (see supplement). Therefore, we utilize a multimodal large language model (MLLM) to provide detailed action captions for the videos. Considering that existing open-source multimodal language models are not as proficient in following instructions as large language models (LLM), we first prompt LLaVA-NeXT-Video-32B-Qwen [66] to provide detailed captions for videos (including foreground, background, main subjects, and action information). Then, we prompt Qwen2 [67] to summarize egocentric action descriptions from the aforementioned captions, with human narrations as the supplementary prompt. Through the combination of MLLM and LLM, our textual descriptions can accurately describe egocentric action while ensuring semantic consistency. We also utilize LLM to analyze the *Nouns* and *Verbs* in each textual description, and classify them into hundreds of action categories (as shown in the Figure 2(a)-(b)). The resulting textual descriptions include actions in household environments, outdoor settings, office activities, sports, and skilled operations, thus covering the majority of scenes encountered in egocentric perspectives.

## 3.2 Data Cleaning Pipeline

The data quality significantly influences the effectiveness of training generative models. Prior works [23, 16, 24] have delved into various cleaning strategies to improve video datasets, focusing on aesthetics, semantic coherence, and optical flow magnitude. Based on these cleaning strategies, this paper presents a specialized cleaning pipeline specifically designed for egocentric scenarios. The pipeline is illustrated in the lower part of Figure 1.

**Text-video Consistency** We utilize CLIP EgoVideo [47] and [68] to evaluate the alignment between textual descriptions and video frames, leveraging EgoVideo's focus on action alignment and CLIP's emphasis on global semantic similarity. In particular, evenly-spaced frames are gathered to calculate the EgoVideo similarity with the text. (refer to Figure 2(c) for Egovideo score distribution). Subsequently, these four frames are separately extracted to calculate CLIP similarity with the corresponding text (see Figure 2(d) for CLIP similarity score distribution).

**Frame-frame Consistency** The higher the semantic consistency between video frames, the more conducive it is to generative training. To analyze this relationship, we uniformly extract three frames alongside the first frame to compute frame CLIP similarity. The distribution of semantic consistency is illustrated in Figure 2(e).

**Motion Smoothness** Excessive egocentric motion can lead to video fluctuations, which is detrimental to training visual generation models. To address this issue, we propose measuring the degree of translation variation $\frac{1}{T}\sum_{t=0}^{T-1}(Tr_t - \overline{Tr})^2$ and rotation variation $\frac{1}{T}\sum_{t=0}^{T-1}(Ro_t - \overline{Ro})^2$ to quantify motion smoothness, where $Tr$ and $Ro$ are translation and rotation measured in Sec. 3.1 (see motion smoothness distribution in Figure 2(f)).

**Motion Strength** A typical approach to describe video motion strength is optical flow [69]. Therefore, we first represent video motion by averaging global optical flow (see motion strength distribution in Figure 2(g)), we additionally calculate the *five-point* optical flow, which includes the proportion of optical flow score across pixel intervals: 0–4, 4–8, 8–12, 12–16, and above 16 (more details see supplement). This method offers a multi-faceted perspective on motion strength, addressing both the movement of small foreground objects and the overall camera motion.

**Clarity Assessment** For egocentric scenes, clarity and realism are paramount. Therefore, instead of relying on CLIP for aesthetic scoring [70], we apply DOVER [71] to assess video clarity (refer to Figure 2 for DOVER score distribution), prioritizing visual sharpness and detail in our dataset.

Based on the cleansing metadata, we vary thresholds to filter and obtain high-quality training data. Specifically, experiments are conducted in Sec. 5.2 to explore the effects of three mainstream

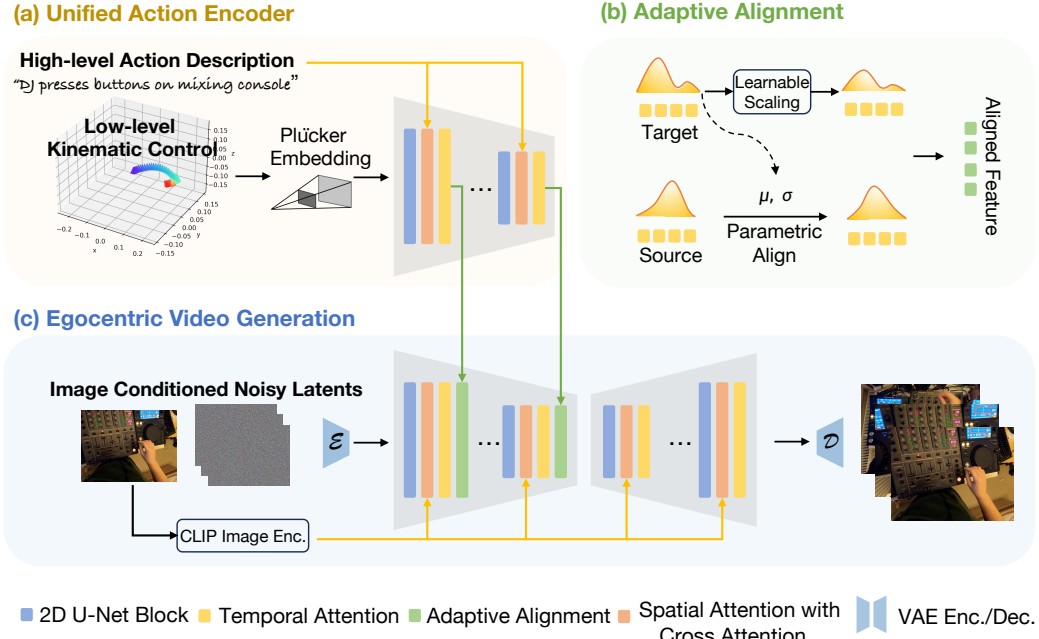

Figure 3: The overall framework of *EgoDreamer*. *EgoDreamer* introduces (a) the Unified Action Encoder to embed different action inputs, and it utilizes (b) the Adaptive Alignment to integrate action conditions into egocentric video generation branch (c).

cleaning strategies on egocentric video generation training. Additionally, given the significance of data cleaning strategies in training video generation models [23, 16, 24], and the substantial computational cost—**thousands of GPU days**—to annotate and clean millions of videos, we release all annotation and cleansing metadata to encourage community research into the impact of various cleaning strategies on egocentric video training.

## 4 EgoDreamer

In the context of ego-centric world simulators, action-driven video generation is paramount. However, existing action-driven video generation approaches [54, 55, 56, 57, 58, 59, 60, 61] primarily focus on camera movements within static scenes, making it challenging to model complex ego-motion. Therefore, we propose *EgoDreamer*, which can produce egocentric videos driven simultaneously by high-level action descriptions and low-level kinematic control. As illustrated in Figure 3, *EgoDreamer* adopts a similar architecture of [17] to enable image-conditioned video generation. Besides, *EgoDreamer* features two key innovations: (1) It introduces a Unified Action Encoder (UAE) that embeds two distinct action inputs, allowing for a more nuanced representation of ego movements. (2) It leverages Adaptive Alignment (AA) that encapsulates multi-scale control signals in the parametric alignment perspective, enhancing the action control efficacy.

**Unified Action Encoder.** In this framework, the UAE simultaneously encodes both low-level and high-level actions. Specifically, it first utilizes Plücker embedding [57, 72] to encode kinematic signals:

$$\mathbf{p}_{u,v} = (\mathbf{t} \times \mathbf{d}_{u,v}, \mathbf{d}_{u,v}), \tag{4}$$

$$\mathbf{d}_{u,v} = \mathbf{R}\mathbf{K}^{-1}[u, v, 1]^T + \mathbf{t}, \tag{5}$$

where $\mathbf{R}$ and $\mathbf{t}$ is the rotation matrix and translation vector, $\mathbf{K}$ is the intrinsic matrix, and $\mathbf{p}_{u,v}$ is the Plücker embedding at pixel $(u, v)$. Then, low-level signal $\mathbf{p}$ is encoded through a series of U-Net blocks, while a high-level action description $d$ is simultaneously embedded via CLIP [68] and cross-attention mechanisms. The action output $A$ of one U-Net block can be formulated as:

$$A = \mathcal{F}_t(\mathcal{F}_c(\mathcal{F}_s(\mathcal{F}_{\text{conv}}(\mathbf{p})), \text{CLIP}(d))), \tag{6}$$

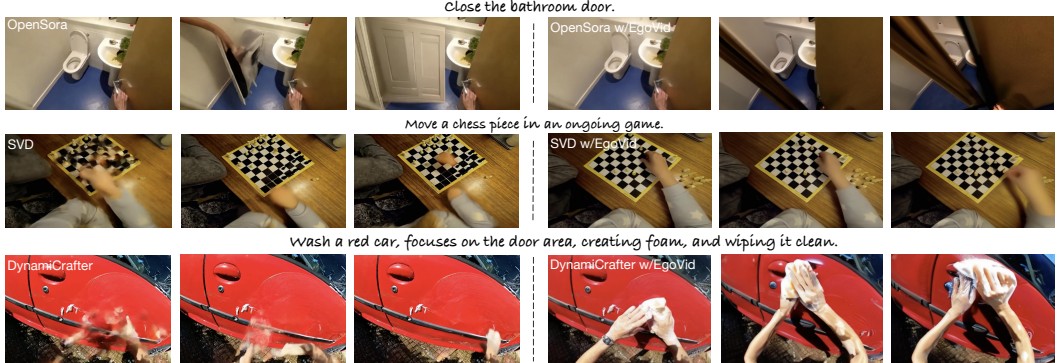

Figure 4: Visualizations demonstrate that *EgoVid*-fintuned baselines (OpenSora [33], SVD [16], DynamiCrafter [17]) generate egocentric videos with stronger frame-consistency and better semantic-alignment.

| Method | w. *EgoVid* | CD-FVD ↓ | Semantic Consistency ↑ | Action Consistency ↑ | Clarity Score ↑ | Motion Smoothness ↑ | Motion Strength ↑ |
|---|---|---|---|---|---|---|---|
| SVD [16] | ✗ | 591.61 | 0.258 | 0.465 | 0.479 | 0.971 | 18.897 |
| SVD [16] | ✓ | **548.32** | **0.266** | **0.471** | **0.485** | **0.974** | **21.032** |
| DynamiCrafter [17] | ✗ | 243.63 | 0.257 | 0.481 | 0.473 | 0.986 | 9.357 |
| DynamiCrafter [17] | ✓ | **236.82** | **0.265** | **0.494** | **0.483** | **0.987** | **18.329** |
| OpenSora [33] | ✗ | 809.46 | 0.260 | 0.489 | 0.520 | 0.983 | 7.608 |
| OpenSora [33] | ✓ | **718.32** | **0.266** | **0.494** | **0.528** | **0.986** | **15.871** |

Table 2: *EgoVid* significantly enhances egocentric video generation. Experimental results demonstrate that training with *EgoVid* improves performance across all three baselines on six metrics.

where $\mathcal{F}_t$ is the temporal self-attention, $\mathcal{F}_c$ is the cross-attention, $\mathcal{F}_s$ is the spatial self-attention, $\mathcal{F}_{\mathrm{conv}}$ is the 2D convolution block. Notably, previous methods [56, 57] encode text and kinematics separately, ignoring that low-level kinematics and high-level action descriptions are coupled. In contrast, the proposed UAE focuses on modeling the relationship between different action inputs, thus the generated action control signals capture both camera movements and complex egocentric dynamics (e.g., hand interactions).

**Adaptive Alignment.** Based on the multi-scale U-Net architecture, the UAE outputs multi-scale $\{A_i\}_{i=0}^3$. Then *EgoDreamer* encapsulates control signals in the perspective of parametric alignment:

$$L_i = \alpha L_i + \frac{A_i - \mu_L}{\sigma_L}, \tag{7}$$

where $L_i$ is the output of one U-Net block in the main Diffusion branch, $\alpha$ is a learnable parameter, $\mu_L, \sigma_L$ are the mean and standard deviation of $L_i$. The introduced AA module is inspired by cross normalization [73] and applies it to multi-scale U-Net feature alignment. Compared to ControlNet's zero-initialization [74], our method achieves better control effectiveness.

## 5 Experiment

### 5.1 Experiment Details

**Dataset.** The proposed *EgoVid-5M* dataset is partitioned as the training set and the validation set. For the validation set, we select samples with high text-video semantic consistency, moderate video motion, high video clarity, and diverse scene coverage including household environments, outdoor settings, office activities, sports, and skilled operations. This resulted in a final validation set *EgoVid-val* with 1.2K samples, with a training set *EgoVid-train* with 4.9M samples. Notably, due to the known issue in Ego4D IMU annotation*, we annotate kinematic controls for 65K video samples with accurate IMU data. The annotated subset *EgoVid-65K* is ∼ 5× larger than the current largest kinematic annotation dataset [56], which is utilized further to train the ability of kinematic control video generation.

---

*https://ego4d-data.org/docs/data/imu/

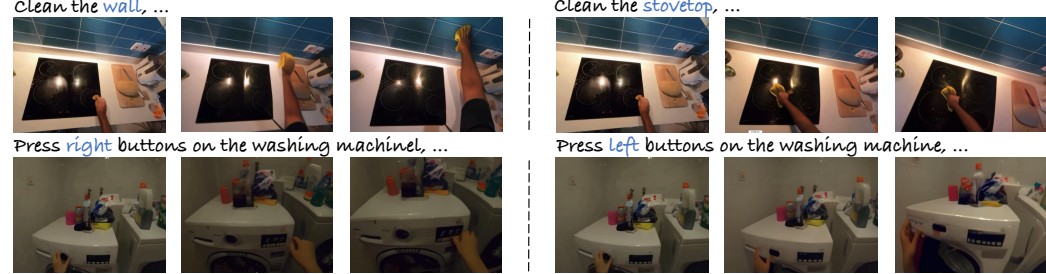

Figure 5: Visualizations show that *EgoDreamer* can realize distinct action controls based on different text descriptions.

**Training** We validate the effectiveness of our *EgoVid-5M* using video diffusion baselines with different architectures, including U-Net (SVD [16] and DynamiCrafter [17]), and DiT (OpenSora [33]). Building upon these pre-trained models, we employ a continuous training approach to train 480p videos for enhanced training efficiency. For *EgoDreamer*, we first initialize it with pre-trained weights [17], then *EgoDreamer* are further trained on *EgoVid* to adapt to egocentric scenes. Finally, we finetune the proposed UAE and AA using *EgoVid-65K*. All experiments are conducted on NVIDIA A800 GPUs. For additional training details, please refer to the supplementary materials.

**Evaluation.** We adopt a set of metrics from AIGCBench [75] and VBench [76] to assess the quality of the generated egocentric videos. Specifically, our evaluation metrics utilize the CD-FVD [77] for spatial and temporal quality, the CLIP [68] for semantic consistency, the EgoVideo [47] for action consistency, the DOVER [71] for clarity score, frame interpolation model [78] for motion smoothness, and RAFT [69] for motion strength. Additionally, following [56, 57], we assess kinematic control consistency using translation error and rotation error, which measures the difference between COLMAP poses and the ground truth poses in the canonical space [56]. The specific calculations for each metric are detailed in the supplement.

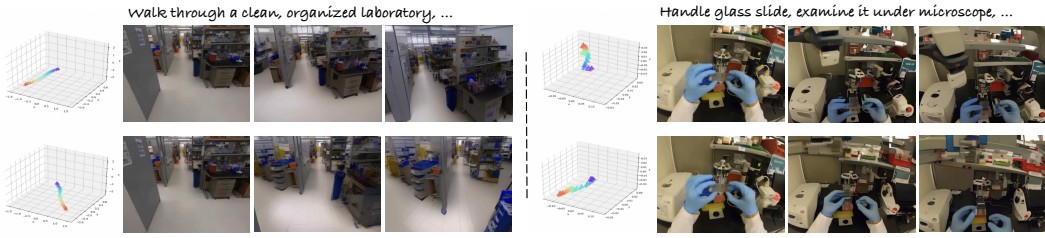

Figure 6: Visualizations demonstrate that *EgoDreamer* can generate various egocentric videos based on different low-level commands, where the left-side poses denotes the kinematic movements in 3D space (from blue to red).

| w. EgoVid | ControlNet | ControlNeXt | AA | UAE | CD-FVD ↓ | Semantic Consistency ↑ | Action Consistency ↑ | Rot Err ↓ | Trans Err ↓ |
|---|---|---|---|---|---|---|---|---|---|
| | ✓ | | | | 241.90 | 0.263 | 0.490 | 5.32 | 9.27 |
| ✓ | ✓ | | | | 238.87 | 0.266 | 0.493 | 4.01 | 8.66 |
| ✓ | ✓ | | | ✓ | 239.01 | 0.268 | 0.494 | 3.58 | 8.41 |
| ✓ | | ✓ | | ✓ | 234.13 | **0.269** | 0.497 | 3.59 | 7.93 |
| ✓ | | | ✓ | ✓ | **229.82** | 0.268 | **0.498** | **3.28** | **7.62** |

Table 3: Ablation study on training strategy and different components of *EgoDreamer*.

Next, we verify the impact of different data cleaning strategies on egocentric video generation. Subsequently, we substantiate, quantitatively and qualitatively, that the proposed *EgoVid* can enhance various baselines' egocentric video generation capabilities. Finally, experiments are conducted to demonstrate that the proposed *EgoDreamer* can generate egocentric videos under the control of both action descriptions and kinematic signals.

## 5.2 Data Cleaning Strategy Comparison

In this subsection, we employ the state-of-the-art video diffusion model DynamiCrafter [17] as the baseline, which is trained on the *Image+Text-to-Video* task to evaluate various data cleaning strategies.

**Strategy-1.** This strategy focuses on ensuring text-frame consistency (with $\text{CLIP}_{TF} \geq 0.275$) and frame-frame consistency ($\text{CLIP}_{FF} \geq 0.8$). Additionally, we retained videos with an average optical flow $\geq 3$ and a DOVER score $\geq 0.3$. This process yielded a subset *EgoVid-1M-1*. DynamiCrafter is finetuned for one epoch using this subset. As illustrated in the supplement, this model achieved the highest semantic consistency metrics. However, the stringent criteria for both text-frame and frame-frame consistency favored the retention of videos with slow motion. Consequently, the motion strength of the generated videos falls below the baseline, which is not desirable for effective video generation.

**Strategy-2.** The thresholds for text-frame consistency and frame-frame consistency are relaxed ($\text{CLIP}_{TF} \geq 0.27$, $\text{CLIP}_{FF} \geq 0.75$). Besides, we retain videos with an average optical flow between 3 and 40, and those with a DOVER score $\geq 0.3$. This strategy results in a subset *EgoVid-1M-2*. Upon finetuning DynamiCrafter for one full epoch, as shown in the supplement, we observe a significant improvement in the motion strength. However, the accelerated motion introduces artifacts, leading to visual fragmentation. Consequently, this negatively impacts the text-frame semantic consistency, resulting in scores below the baseline.

**Strategy-3.** we further relax the thresholds for text-frame consistency ($\text{CLIP}_{TF} \geq 0.26$) and frame-frame consistency ($\text{CLIP}_{FF} \geq 0.7$), while introducing an action consistency constraint (EgoVideo score $\geq 0.22$). Videos are retained with an average optical flow between 3 and 35, as well as those with a DOVER score $\geq 0.3$. Notably, as mentioned in Sec. 3.2, we also retain videos with average optical flow values below 3, provided that the proportion of optical flow ($\geq 12$ pixels) is greater than 3%. This resulted in the *EgoVid-1M-3* subset. Compared to the previous two strategies, the model finetuned on *EgoVid-1M-3* effectively enhances both semantic and action consistency while ensuring moderate motion strength, achieving the best CD-FVD score. Furthermore, the *5-point* optical flow filtering method allowed for a focus on local motion scenarios. As illustrated visualizations in supplement, strategy-3 accurately models intricate hand movements, in contrast to the stationary visuals of strategy-1 and the exaggerated motion of strategy-2.

## 5.3 Enhancement in Egocentric Video Generation

In this subsection, experiments are conducted to verify that the proposed *EgoVid* enhances the egocentric video generation capabilities of various baselines. Specifically, SVD [16], DynamiCrafter [17], and OpenSora [33] are selected as baselines, which are initialized with their original weights, and then we employ *EgoVid-1M-3* for finetuning. For training efficiency and fair comparison, we resize all input video to 480p and focus exclusively on the *Image+Text-to-Video* tasks. As shown in Table 2, the experiment results demonstrate that training with *EgoVid* improves performance across all three baselines on six different metrics. Specifically, the *EgoVid* finetuning significantly enhances the motion strength of egocentric videos while also improving consistency in text-video alignment, action-video alignment, and overall image clarity. Consequently, the CD-FVD metric shows a notable improvement. Additionally, we conduct a visualization comparison of different baselines before and after finetuning, as illustrated in Figure 4. Prior to *EgoVid* finetuning, various baselines exhibit issues such as frame fragmentation and distortion in egocentric scenarios (e.g., appearance of incongruous objects and hand fragmentation). This underscores the inadequacy of most existing video generation models in egocentric contexts. However, after the *EgoVid* finetuning, the generated videos not only achieve superior alignment with text prompts, but also exhibit enhancement in visual quality.

## 5.4 EgoDreamer Experiments

In this subsection, we conduct experiments to demonstrate that *EgoDreamer* can generate egocentric videos under the control of both action descriptions and kinematic signals. Additionally, the efficacy of the proposed UAE and AA modules will be validated. In our experiments, we initialize *EgoDreamer* using weights from [17]. The results are presented in Table 3. In Row-1, the low-level kinematic control signals are integrated via ControlNet [77], which resembles [56, 57]. Row-2 utilizes *EgoVid-1M-3* to pre-train the model. Compared with Row-1, results indicate significant improvements across five metrics after *EgoVid-1M-3* finetuning. In Row-3, we further introduce the UAE module to strengthen the association between low-level kinematic control and high-level action descriptions.

The experimental results indicate that this enhancement further improves action alignment and reduces the deviation in low-level kinematic control compared to Row-2. In Row-4 and Row-5, we replace the ControlNet with ControlNext [73] and the AA module. The results reveal that the AA module exhibits superior performance compared to both ControlNet and ControlNext, as it facilitates learnable parameterized alignment from a multi-scale perspective. Finally, we visualize videos generated by *EgoDreamer*, as depicted in Figure 5. Under initial frame conditions, varying the input text descriptions enables *EgoDreamer* to realize distinct action controls. Furthermore, as illustrated in Figure 6, with the same initial frame, the model can generate videos that incorporate a composite of multiple low-level kinematic controls. Notably, *EgoDreamer* to produce videos with meter-level movements (e.g., walking) and centimeter-level nuanced movements (e.g., intricate hand actions in a laboratory environment). Additional visualizations can be found in the supplement.

# 6 Conclusion and Limitations

**Conclusion** We present *EgoVid-5M*, the first large-scale, high-quality dataset tailored for egocentric video generation, containing 5 million clips with fine-grained action annotations. To ensure data quality, we introduce a robust cleaning pipeline that enforces temporal consistency and motion coherence. We also propose *EgoDreamer*, a generative framework that synthesizes egocentric videos conditioned on both action labels and kinematic signals. We hope *EgoVid-5M* will foster further research in egocentric video generation and benefit applications in VR, AR, and immersive simulation.

**Limitations** Despite its scale and quality, *EgoVid-5M* focuses primarily on short clips and pre-defined action categories, which may limit generalization to long-horizon tasks or open-ended activity understanding. Additionally, while *EgoDreamer* integrates kinematic control, its current form assumes pre-specified trajectories rather than learning them jointly, leaving room for future improvements in closed-loop control and interactive generation.

# 7 Acknowledgment

This work was supported by Alibaba Research Intern Program.

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

## A   Technical Appendices and Supplementary Material

Technical appendices with additional results, figures, graphs and proofs may be submitted with the paper submission before the full submission deadline (see above), or as a separate PDF in the ZIP file below before the supplementary material deadline. There is no page limit for the technical appendices.

