# OpenReview forum: "EgoVid-5M: A Large-Scale Video-Action Dataset for Egocentric Videos Generation"
_NeurIPS.cc/2025/Datasets_and_Benchmarks_Track — NeurIPS 2025 Datasets and Benchmarks Track poster_

### Official Review · Reviewer_CjG8 · 2025-06-19

**Rating:** 4
**Confidence:** 4

**Summary:**

This paper presents EgoVid-5M, a large-scale dataset for egocentric video generation, featuring 5 million video clips with detailed action annotations. It also introduces EgoDreamer, a model that generates egocentric videos using both action descriptions and kinematic control.

**Additional Feedback:**

At this stage, my main concern is that the value of the proposed dataset seems limited. However, overall, it should still meet the acceptance bar.

**Dataset Code Accessibility:**

Yes

**Ethical Considerations:**

No, there are no or only very minor ethics concerns

**Final Justification:**

Since all of my concerns have been resolved, I am inclined to accept the paper. The authors should include the additional discussions in the revised version as they have promised.

**Limitations Weaknesses:**

- The proposed dataset appears to be a collection of video clips annotated with camera pose, but does the trained EgoDreamer allow for both quantitative and qualitative comparisons with camera control methods? This part seems to be missing.
- Given that the dataset consists of short video clips, this significantly limits its practical applications. In the introduction, the authors mention potential applications like autonomous driving and gaming, but both of these fields typically require videos of a certain length, or even long videos. This limitation reduces the overall value of the dataset. I'm curious about what other practical applications this video clip dataset might be useful for, besides training models for camera control video generation. I'm not convinced by the claim like "future work will address this limitation."

**Strengths Contributions:**

- The dataset is rich and contains a sufficient number of video clips, with effective action annotations and a sound data cleaning approach.
- EgoDreamer has been shown in experiments to effectively generate videos with camera control.

---

> ### Author Rebuttal · Authors · 2025-07-30
>
> We thank the reviewer for their thoughtful feedback. We appreciate the recognition of our dataset’s scale, annotation quality, and the effectiveness of EgoDreamer in generating egocentric videos with camera control. Below, we address the reviewer’s concerns in detail.
> - Concern 1:  Comparison with Camera Control Methods
>   - Response 1: We appreciate the reviewer’s comment. Since our task focuses on egocentric video generation, existing camera control methods perform poorly under our setup due to significant domain differences in motion patterns and viewpoint dynamics. To enable a fair comparison, we adapt CameraCtrl [1] framework to our task by retraining it on EgoVid-5M using the same action and pose conditioning. As detailed in Section 3.2 (Line 297), the experimental setup in Table 3 (Row 2 vs. Row 1) ensures a comparable configuration. The results show that the adapted CameraCtrl benefits significantly from training on our EgoVid-5M dataset: rotation and translation errors are reduced by 24.6% and 6.6%, respectively, demonstrating the effectiveness and value of EgoVid-5M for learning egocentric camera dynamics. Furthermore, as shown in Table 3 (Row 5), our proposed EgoDreamer significantly outperforms the adapted CameraCtrl (Row 2), achieving an additional 18.2% reduction in rotation error and 12.0% reduction in translation error, while also producing more temporally coherent and visually plausible egocentric videos. This highlights the superiority of our model design in capturing the coupled dynamics of human action and camera motion. In the revised manuscript, we will strengthen the discussion of these comparisons, include additional visual comparisons to further illustrate the advantages of both EgoVid-5M and EgoDreamer in the context of camera-controlled egocentric video generation.
> - Concern 2:  Limitations of Short Video Clips and Practical Applicability
>   - Response 2: We  thank the reviewer for the thoughtful feedback. The proposed EgoVid dataset consists of video clips centered on individual egocentric actions, with each clip typically lasting around 5 seconds. We believe EgoVid-5M fills a critical gap in the field of generative modeling for egocentric vision. Prior to this work, no large-scale, action-annotated dataset was available to support training and evaluation in this domain. The dataset offers several key advantages:
>     - Fine-grained action and camera control modeling: EgoVid-5M enables training and evaluation of models that generate egocentric videos conditioned on specific human actions and poses, facilitating precise control over both content and viewpoint dynamics.
>     - Pretraining foundation models: As demonstrated in Figure 4, existing foundation video generation models perform poorly on egocentric video generation, likely due to their training on egocentric videos. EgoVid-5M can serve as a valuable pretraining resource to enhance the egocentric understanding and generation capabilities of such models.
>     - Supporting embodied AI research: As highlighted in [2–4], data scarcity remains a major bottleneck in developing embodied agents. First-person videos are increasingly used to train Vision-and-Language-Action (VLA) models for robotic manipulation. EgoVid-5M can act as a rich supplementary data source in this context. Moreover, video generation models trained on EgoVid (e.g., EgoDreamer) can further synthesize diverse, action-conditioned egocentric videos, enabling data augmentation for embodied learning.
>
>   We appreciate the reviewer’s valuable comments. In the revised manuscript, we will expand the discussion of these points to better highlight the broader impact and utility of the EgoVid-5M dataset.
>
> [1] Hao He, Yinghao Xu, Yuwei Guo, Gordon Wetzstein, Bo Dai, Hongsheng Li, and Ceyuan Yang. Cameractrl: Enabling camera control for text-to-video generation. arXiv preprint arXiv:2404.02101, 2024.
>
> [2] Ruihan Yang, Qinxi Yu, Yecheng Wu, Rui Yan, Borui Li, An-Chieh Cheng, Xueyan Zou et al.  EgoVLA: Learning Vision-Language-Action Models from Egocentric Human Videos.  arXiv preprint arXiv:2507.12440, 2025.
>
> [3] Qingwen Bu, Yanting Yang, Jisong Cai, Shenyuan Gao, Guanghui Ren, Maoqing Yao, Ping Luo, and Hongyang Li. Univla: Learning to act anywhere with task-centric latent actions. arXiv preprint arXiv:2505.06111, 2025.
>
> [4] Chilam Cheang, Sijin Chen, Zhongren Cui, Yingdong Hu, Liqun Huang, Tao Kong, Hang Li et al. GR-3 Technical Report. arXiv preprint arXiv:2507.15493, 2025.
>
> Thanks again for the professional, detailed, and valuable reviews! We have done our best to address each of the above concerns and hope our response can resolve them. Please let us know if there are any other questions. We will actively join the discussion until the end of the rebuttal period.

---

> > ### Comment · Reviewer_CjG8 · 2025-08-02
> >
> > Thank you to the authors for their response. My concerns have been largely addressed, and I hope the authors will incorporate the promised changes in the revised version.

---

> > > ### Author Response · Authors · 2025-08-02
> > >
> > > Thank you for your feedback. We will carefully revise the manuscript and ensure all promised changes are incorporated in the revised version.

---

### Official Review · Reviewer_sCgo · 2025-06-25

**Rating:** 5
**Confidence:** 4

**Summary:**

This paper proposes EgoVid-5M, a large-scale dataset comprising 5 million high quality egocentric video clips that cover diverse scenes. The dataset is originated from Ego4D, where a data cleaning pipeline is applied. The dataset also comes with detailed annotations, including kinematic control and action descriptions. Moreover, the authors propose EgoDreamer, a conditional egocentric video generation method, to showcase that EgoVid-5M can be used to train generative models.

**Dataset Code Accessibility:**

Partly

**Dataset Code Comments:**

The dataset is accessible, but I could not find the source code to reproduce the experiments.

**Ethical Considerations:**

No, there are no or only very minor ethics concerns

**Final Justification:**

All my concerns have been addressed, so I recommend accepting this paper. The authors should include all the promised updates in the final revision.

**Limitations Weaknesses:**

- The kinematic control is only annotated for a subset of 65k video clips as stated in Section 5.1. However, this information is not mentioned in previous sections, which may mislead the readers to believe that the kinematic control is available for all the 5M clips at first glance. This is an important piece of information that needs to be made very clear upfront in the Abstract and the Introduction section.
- Most of the visual results for EgoDreamer are indoor scenes. Since the conditional generation part is trained on the 65k subset, does it mean that the 65k subset is highly biased towards indoor scenes? It would be great if the authors can provide the statistics for the 65k subset similar to what is done for the full dataset.
- This paper is well-organized and easy to follow, but the figures are quite hard to read, especially the text in the figures. The authors may need to increase the font size in the figures, or to redesign the figure layout so the ideas can be better conveyed.
- As the authors acknowledge in Section 6, the focus on short clips and pre-defined actions may restrict the application of the proposed dataset. It would be valuable if the authors can discuss the potential challenges of expanding the dataset by applying the same data processing pipeline to longer videos and broader scopes.

**Strengths Contributions:**

- Although the raw data are from the Ego4D dataset, this paper puts significant effort into data cleaning to ensure temporal consistency and motion coherence. With the annotation of kinematic control and action descriptions, the proposed dataset is suitable for training egocentric video generation models driven by both high-level action descriptions and low-level kinematic control.
- This paper proposes EgoDreamer, a novel egocentric video generation method driven by action labels and kinematic signals. The key components like UAE (Unified Action Encoder) and AA (Action Alignment) modules can effectively integrate different data modalities. It is a demonstration of the successful application of the proposed dataset, and can be used as a baseline for future studies on this topic.
- Extensive experiments and insightful analyses are given, including the impact of data cleaning strategies on egocentric video generation, the effectiveness of EgoVid-5M on enhancing existing egocentric video generation methods, and the capability of EgoDreamer on conditional egocentric video generation.

---

> ### Author Rebuttal · Authors · 2025-07-30
>
> We thank the reviewer for the constructive feedback. We appreciate the positive comments regarding our significant data cleaning effort, the novelty of EgoDreamer, and the extensive experiments provided. Below, we address the specific concerns raised:
> - Concern 1: Clarity on Kinematic Control Coverage
>   - Response1 : The coverage of kinematic control annotations (65k clips out of 5M) is crucial information that should be stated upfront. We will revise both the Abstract and the Introduction section to explicitly mention that fine-grained kinematic control annotations are available for a subset of 65k clips, while high-level textual descriptions are available for the full 5M clips.
> - Concern 2: Statistics of the 65k Kinematic Subset
>   - Response 2: Thank you for this insightful question. The 65k kinematic subset includes 34.3% household environments,  23.5% office activities, 21.2% skilled operations and 21% outdoor settings. The subset maintains diversity across various environments and is not exclusively biased towards indoor settings. Furthermore, we will provide the detailed statistics for the 65k kinematic subset in the revised manuscript, following the same format as the full dataset (e.g., Figure 2 in the main paper), to give a clear and comprehensive overview of its composition.
> - Concern 3: Figure Readability
>   - Response 3: We appreciate this feedback. We will increase the font sizes, adjust the figure layouts for better clarity, and ensure all key information is easily legible.
> - Concern 4: Challenges of Expanding to Longer Videos
>   - Response 4: Our data processing pipeline can be extended to longer videos and broader action categories, but several challenges remain. First, longer videos introduce increased complexity in temporal annotation, as identifying precise start and end times of actions becomes more ambiguous. Second, broadening the scope of actions increases the risk of class imbalance, which could affect model performance. Ensuring comprehensive coverage while maintaining well-defined action boundaries would require careful taxonomy design. Nonetheless, we agree that such an extension would enhance the dataset’s applicability. As future work, we plan to explore leveraging action detection and segmentation models, to scale our pipeline efficiently to longer videos and more diverse action categories. We will also conduct ablation studies to assess the impact of video duration and action diversity on model generalization.
> - Concern 5: Code Availability
>   - Response 5: We have updated the HuggingFace repository and the GitHub repository with detailed data annotation code, including high-level caption annotation pipeline (LLava with Qwen), low-level action annotation (VIO with ParticleSfM), and  metadata annotation (DOVER score, EgoVideo score, CLIP score, Optical-flow score).
>
>
> Thanks again for helping us improve the paper, and we hope the response can resolve these concerns! Please let us know if there are any further questions. We will be actively available until the end of the rebuttal period.

---

> > ### Comment · Reviewer_sCgo · 2025-08-03
> >
> > Thanks for the authors' clarification! All my concerns have been resolved.

---

### Official Review · Reviewer_YnB5 · 2025-06-27

**Rating:** 5
**Confidence:** 4

**Summary:**

A high-quality dataset tailored for egocentric video generation containing 5 million clips with fine-grained action annotations.

- Strengths
  - Both quality and quantity of samples look better to previous relative datasets, and it somehow fills the gap in ego-centric video generation;
  - Well accessiblity. It can easily found on huggingface page and  loaded by Python code;

- Weakness
  - Lack of page to display gallery of thumbnails of videos;
  - Lack of multiple sources such as baiduyun, google drive, etc.

It's overall good.

**Dataset Code Accessibility:**

Yes

**Ethical Considerations:**

Yes, there are significant ethics concerns that require review by an ethics expert

**Final Justification:**

# Final Justification for Recommended Score

After reviewing the authors’ final remarks, rebuttal, and discussions with other reviewers and the Area Chair, I recommend a positive score for the EgoVid-5M dataset and EgoDreamer model paper. Below is a simplified summary of resolved and unresolved issues, their weights, and key points for the authors.

## Resolved Issues
1. **Kinematic Annotations**: The authors clarified that kinematic annotations cover a 65k subset with detailed statistics, improving transparency. (*Weight: High*)
2. **Short Clips’ Utility**: Expanded discussion on short clips’ use in pretraining and embodied AI addressed practical value concerns. (*Weight: Moderate*)
3. **Accessibility**: Adding a thumbnail gallery, extra download options, and full code release resolved usability and reproducibility issues. (*Weight: High*)
4. **Ethics**: Commitment to ethical statements and dual-use discussion addressed ethical concerns. (*Weight: Moderate*)

## Unresolved Issues
1. **Evaluation Bias**: The validation set’s construction remains unclear, raising concerns about bias and generalizability of CD-FVD scores. Lack of qualitative evaluation limits result confidence. (*Weight: High*)
2. **CLIP/VIO Limitations**: Acknowledged issues with CLIP embeddings and VIO annotations persist without solutions (e.g., offline bundle adjustment). (*Weight: Moderate*)

## Weighting and Recommendation
- The dataset’s scale (5M clips), quality, and EgoDreamer’s novel dual conditioning are major contributions to egocentric video generation. (*Positive, high weight*)
- Resolved issues on annotations, accessibility, and ethics boost the paper’s impact. (*Positive, high weight*)
- Unresolved evaluation bias tempers enthusiasm but doesn’t negate value. (*Negative, high weight*)
- CLIP/VIO limitations are notable but less critical given the dataset’s utility. (*Negative, moderate weight*)

## Key Points for Authors
1. Clarify validation set construction and add qualitative evaluations to address bias concerns.
2. Explore solutions for CLIP and VIO limitations (e.g., offline bundle adjustment).
3. Ensure promised revisions (thumbnail gallery, downloads, ethics) are implemented.
4. Emphasize EgoVid-5M as a curated Ego4D subset to avoid overstating novelty.

Overall, EgoVid-5M and EgoDreamer offer strong contributions, with evaluation concerns slightly moderating the score.

**Limitations Weaknesses:**

- Weakness
  - Lack of page to display gallery of thumbnails of videos;
  - Lack of multiple sources such as baiduyun, google drive, etc.

**Strengths Contributions:**

- Strengths
  - Both quality and quantity of samples look better to previous relative datasets, and it somehow fills the gap in ego-centric video generation;
  - Well accessiblity. It can easily found on huggingface page and  loaded by Python code;

---

> ### Author Rebuttal · Authors · 2025-07-30
>
> We sincerely thank the reviewer for the positive feedback and the constructive suggestions for improving the accessibility and usability of our EgoVid-5M dataset. We are delighted that the reviewer recognizes the quality and potential impact of our work.
> - Addressing the Suggestions:
>   - Gallery of Thumbnails: We will create a webpage to display thumbnails of the video clips. This will be made available alongside the HuggingFace repository.
>   - Multiple Download Sources: We understand that providing alternative download options can improve accessibility for users worldwide. We will investigate the feasibility of hosting the dataset (or providing mirrors) on platforms like Google Drive and Baidu Netdisk to complement the existing HuggingFace hosting.
> - Regarding Ethical Considerations:
>   - We respectfully clarify that the EgoVid-5M dataset is derived from the publicly available Ego4D [1] dataset, which has undergone its own ethical review and anonymization processes. Our curation and filtering pipeline does not introduce new data collection or processing steps that would inherently raise significant new ethical issues beyond those addressed by Ego4D. Our paper adheres to the ethical guidelines for using this existing dataset. We are committed to responsible data usage and will ensure our release includes appropriate documentation regarding data origin and usage rights.
> We believe these additions and clarifications will further strengthen the accessibility and overall quality of our contribution. Thanks again for the valuable feedback.
>
> [1]  Kristen Grauman, Andrew Westbury, Eugene Byrne, Zachary Chavis, Antonino Furnari, Rohit Girdhar, Jackson Hamburger, Hao Jiang, Miao Liu, Xingyu Liu, et al. Ego4d: Around the world in 3,000 hours of egocentric video. In CVPR, 2022.
>
>
> Thanks again for the professional, detailed, and valuable reviews! We have done our best to address each of the above concerns and hope our response can resolve them. Please let us know if there are any other questions. We will actively join the discussion until the end of the rebuttal period.

---

> > ### Comment · Reviewer_YnB5 · 2025-08-07
> > **Official Comment**
> >
> > Thank you for your thorough response. After carefully reviewing your reply and considering the discussions from other reviewers, I believe the work is strong in both its contributions and ethical standards. As such, I have decided to maintain my positive rating.

---

> ### Comment · Area_Chair_BLrN · 2025-08-05
>
> Dear reviewer,
> please read the other reviews and the author response, and start a discussion with the authors promptly to allow time for an exchange.
> Your AC

---

### Official Review · Reviewer_AQTB · 2025-07-05

**Rating:** 4
**Confidence:** 3

**Summary:**

This paper introduces EgoVid-5M, a large-scale, high-quality dataset curated specifically for egocentric video generation. The dataset contains 5 million 1080p video clips derived from Ego4D, annotated with fine-grained kinematic control using visual-inertial odometry (VIO) and high-level textual descriptions generated via a multimodal LLM pipeline. To demonstrate the utility of the dataset, the authors propose EgoDreamer, a generative model that synthesizes egocentric videos conditioned on both action descriptions and kinematic signals. The dataset and model are evaluated using a comprehensive set of metrics—including CD-FVD, CLIP-based semantic similarity, and optical flow-based motion strength—showing that training with EgoVid-5M improves performance across multiple baselines.

**Additional Feedback:**

Other than concerns mentioned in weaknesses, I would really like to see if the video generation model can be conditioned on the textual description of head trajectory rather than actual trajectory which might not be available at test time. If authors can show how well the model understands - walk straight into the room and turn left followed by sitting on the chair or similar geometric descriptions, that would have additional utility. Also, I wonder how well can the model infer the scale of the trajectory or subtle head motions like nod or head shake (very common in egocentric videos while exploring an environment).

**Dataset Code Accessibility:**

Yes

**Ethical Considerations:**

No, there are no or only very minor ethics concerns

**Final Justification:**

Most of my concerns were addressed. Certain things can be improved in the benchmark such as evaluation of long form video generation, testing how well actions have been completed such as detection of change in object states and improving head pose estimation quality using offline bundle adjustment. Nevertheless, the design decisions are justified and considering the scale of Ego4D the proposed benchmark with manually curated videos for validation and providing intermediate results such as head trajectories+captions can help community to use authors' findings improve egocentric video generation quality.

**Limitations Weaknesses:**

1. While extensive, the dataset is derived from Ego4D, and the contribution primarily lies in the data selection and filtering for training, rather than from collection of novel content. This distinction should be more clearly acknowledged.
2. The video CLIP embeddings may not capture fine-grained visual details, such as subtle differences in object states (e.g., cutting vs. peeling), which can lead to incorrect filtering. This is further compounded by the fact that narrations are automatically generated, introducing additional noise into the filtering pipeline.
3. The kinematic annotations rely on VIO-based pose estimation, which may suffer in accuracy under fast motion or sensor noise. An offline bundle adjustment method that jointly optimizes trajectory and camera intrinsics/extrinsics might offer more precise supervision.
4. The majority of evaluation is based on CD-FVD scores on a hand-curated validation set derived from Ego4D. However, the construction of this validation dataset is not clearly explained. If it was sampled using the same heuristics used during training data filtering (e.g., optical flow, motion smoothness, and CLIP-based text-video similarity), it may lead to distributional bias. As a result, the reported performance gains may not generalize beyond this curated subset, and the conclusions drawn could change significantly under different evaluation conditions. Authors should consider qualitative evaluation as well for addressing their design choices and/or sampling strategies.

**Strengths Contributions:**

1. The paper is clearly written and well-structured, making it easy to follow.
2. A new generative model, EgoDreamer, is proposed, which conditions on both high-level action descriptions and low-level kinematic trajectories, enabling fine-grained control over the generated egocentric videos.
3. The proposed data cleaning pipeline is automated and well-justified, removing noise via text-video semantic alignment, motion smoothness, and clarity scores.

---

> ### Author Rebuttal · Authors · 2025-07-30
>
> We thank the reviewer for the thorough and constructive feedback on our work. We appreciate the positive comments regarding the clarity of our paper, the novelty of EgoDreamer, and the robustness of our data cleaning pipeline. We are also grateful for the insightful suggestions for improvement. Below, we address the specific concerns raised:
> - Concern 1: Contribution Clarity
>   - Response 1: We agree with the reviewer's point. Our intention was indeed to highlight the significant curation and filtering effort. However, we acknowledge that the paper could have been clearer in emphasizing that the core contribution lies in the novel processing, filtering, and annotation of existing Ego4D [1] data, rather than collecting entirely new raw footage. We will revise the Introduction and Related Works sections to explicitly state this distinction.
> - Concern 2: Clarifying the Quality of Text Annotation and CLIP Embedding
>   - Response 2: We appreciate this important concern regarding potential hallucinations in VLM and CLIP models. To mitigate this issue and enhance the quality of our text annotations, we employed a dual-annotation strategy: in addition to generating detailed captions using a multimodal LLM, we also leveraged the real human narrations sourced from Ego4D [1] (as described in Lines 140-144) to reduce hallucinations in the LLM-generated annotations. As shown in Figure 5 of the supplementary material, the semantic similarity of our generated captions is approximately 30% higher than that of the original human narrations from Ego4D. Furthermore, we utilized EgoVideo-CLIP [2], a CLIP model fine-tuned on human action data, to evaluate the semantic alignment between textual descriptions and video frames. Compared to standard CLIP model [3], EgoVideo-CLIP demonstrates superior capability in distinguishing subtle differences in egocentric actions, thereby providing more accurate filtering signals. We will elaborate on these points in the revised manuscript to better clarify the robustness of our annotation and filtering process.
> - Concern 3: Kinematic Annotation Accuracy
>   - Response 3: We appreciate this technical suggestion. The VIO can be susceptible to errors under challenging conditions. Our choice of a real-time VIO system was primarily driven by the need to process the vast amount of Ego4D data efficiently. We will discuss this limitation in the revised text and acknowledge that more sophisticated offline methods, like bundle adjustment, could potentially yield higher accuracy annotations for future dataset iterations or specialized subsets. Nevertheless, we believe the current kinematic annotations provide valuable control signals for video generation, as evidenced by EgoDreamer's performance.
> - Concern 4: Validation Set Construction & Qualitative Evaluation
>   - Response 4 (Validation Set Construction):  For the validation set, we constructed it through a combination of data distribution statistics and manual curation. Specifically, we selected samples with high text-video semantic consistency (CLIP_TF ≥ 0.3, CLIP_FF ≥ 0.7, EgoVideo score ≥ 0.25), moderate video motion (optical flow between 3 and 35), and high video clarity (DOVER score ≥ 0.3). Furthermore, we performed manual screening to ensure the videos were free from sudden jitters or other noise artifacts. This process resulted in diverse scene coverage, including household environments (31%), outdoor settings (25%), office activities (20%), sports (15%), and skilled operations (9%). The selection criteria for the evaluation set are stricter than those for the training set and explicitly designed to ensure diversity and quality. This rigorous selection aims to provide a robust benchmark where the generated videos can be meaningfully compared against high-quality real samples. We will revise this detailed construction process to avoid any misunderstanding.
>   - Response 4 (Qualitative Evaluation): Beyond the eight quantitative evaluation metrics used to validate the effectiveness of our method, we have also employed extensive qualitative evaluations. These include visual comparisons shown in the main paper (Figures 4-6) and supplementary video files, which demonstrate the quality of both the EgoVid dataset and the EgoDreamer model. We will include additional high-quality visual results (both generated and real) in the revision to further complement our quantitative metrics and provide a more holistic view of our design choices and sampling strategies.
> - Additional Feedback: Conditioning on Textual Trajectory Description & Subtle Motions
>   - Response (Conditioning on Textual Trajectory Description): We appreciate this insightful suggestion. EgoDreamer is indeed capable of conditioning on high-level textual descriptions, enabling the generation of videos based on action descriptions such as "walk into the room" or "sitting on the couch," as demonstrated in Figure 6 of the supplementary material. However, the generation of continuous, long-duration actions is beyond the current scope of this work and represents a promising direction for future research. We believe our EgoVid-5M dataset, with its large scale and diverse action annotations, can effectively support the training of models for such long-sequence video generation tasks.
>   - Response (Subtle Motions): Furthermore, subtle head motions like nodding or shaking can also be captured through textual descriptions. To enhance the precision of action control, our model can leverage low-level kinematic control to accurately infer trajectory scales. This is made possible because the kinematic annotations in EgoVid-5M are derived using IMU signals, which provide metrically scaled trajectory information (see Equation 1), offering fine-grained control over the generated movements.
>
> We believe these revisions and additions will significantly strengthen our paper. Thanks again for the valuable feedback.
>
> [1]  Kristen Grauman, Andrew Westbury, Eugene Byrne, Zachary Chavis, Antonino Furnari, Rohit Girdhar, Jackson Hamburger, Hao Jiang, Miao Liu, Xingyu Liu, et al. Ego4d: Around the world in 3,000 hours of egocentric video. In CVPR, 2022.
>
> [2] Baoqi Pei, Guo Chen, Jilan Xu, Yuping He, Yicheng Liu, Kanghua Pan, Yifei Huang, Yali Wang, Tong Lu, Limin Wang, et al. Egovideo: Exploring egocentric foundation model and downstream adaptation. arXiv preprint arXiv:2406.18070, 2024.
>
> [3] Radford, Alec, Jong Wook Kim, Chris Hallacy, Aditya Ramesh, Gabriel Goh, Sandhini Agarwal, Girish Sastry et al. "Learning transferable visual models from natural language supervision." In ICML, 2021.
>
>
> Thanks again for helping us improve the paper, and we hope the response can resolve these concerns! Please let us know if there are any further questions. We will be actively available until the end of the rebuttal period.

---

### Note · Authors · 2025-08-13

We would like to thank all reviewers, area chairs for their constructive feedback on our work, and we appreciate the overall positive recognition of our contributions.

Our paper introduces EgoVid-5M, a large-scale, high-quality dataset curated for egocentric video generation, comprising 5 million video clips with fine-grained action annotations and kinematic control signals derived from Ego4D. To demonstrate its utility, we further propose EgoDreamer, a novel generative model that enables precise control over egocentric video synthesis by conditioning on both high-level textual descriptions and low-level kinematic trajectories. We believe this work fills a critical gap in the field by providing a resource for training and evaluating egocentric generative models, while also advancing model design for embodied and first-person vision tasks.

We are pleased that the reviewers have acknowledged the strengths of our work across multiple dimensions:

- Reviewer YnB5 recognized the dataset’s superior quality, noting it “fills the gap in egocentric video generation,” and affirmed its high impact with a top rating.
- Reviewer sCgo emphasized the significant effort in data curation, the effectiveness of the UAE and AA modules in EgoDreamer, and the value of extensive experimental analysis.
- Reviewer AQTB highlighted that the paper is “clearly written and well-structured,” praised the novelty of EgoDreamer’s dual conditioning mechanism, and commended our robust data cleaning pipeline.
- Reviewer CjG8 acknowledged the richness of the dataset and the effectiveness of EgoDreamer in generating videos with camera control.

We have carefully addressed all concerns raised during the review process:

- We clarified the scope of kinematic annotations (65k subset) in the abstract and introduction.
- We provided detailed statistics for the kinematic subset.
- We expanded discussion on the practical applicability of short clips, including use cases in pretraining foundation models and embodied AI.
- We strengthened comparisons with camera control baselines.
- We committed to adding ethical statements and discussing the dual-use potential of generative egocentric video models.
- We improved dataset accessibility by committing to a thumbnail gallery and additional download options, and have released full code for data processing and model reproduction.

All promised revisions will be incorporated into the final version.

---

### Decision · Program_Chairs · 2025-09-18

**Decision:**

Accept (poster)

**Comment:**

The paper introduced EgoVid-5M, a large-scale curated dataset of over 5 million annotated clips for video generation, along with EgoDreamer, a model capable of generating egocentric video guided by both action descriptions and kinematic signals. The paper received positive reviews, with reviewers appreciating the dataset quality, the data curation pipeline and the effectiveness of EgoDreamer. The reviewers appreciated the rebuttal which resolved most of the concerns and the ethical considerations. The AC, given the positive recommendation, the many contributions and the importance of this work for the community (as egocentric video generation is a hot topic in computer vision) recommends acceptance.